# Coarse-to-Fine Semi-Structured Pruning of Graph Convolutional Networks for Skeleton-based Recognition

**Hichem Sahbi** [1]

## Abstract

Deep neural networks (DNNs) are nowadays witnessing a major success in solving many pattern recognition tasks including skeleton-based classification. The deployment of DNNs on edge-devices, endowed with limited time and memory resources, requires designing lightweight and efficient variants of these networks. Pruning is one of the lightweight network design techniques that operate by removing unnecessary network parts, in a structured or an unstructured manner, including individual weights, neurons or even entire channels. Nonetheless, structured and unstructured pruning methods, when applied separately, may either be inefficient or ineffective.

In this paper, we devise a novel semi-structured method that discards the downsides of structured and unstructured pruning while gathering their upsides to some extent. The proposed solution is based on a differentiable cascaded parametrization which combines (i) a band-stop mechanism that prunes weights depending on their magnitudes, (ii) a weight-sharing parametrization that prunes connections either individually or group-wise, and (iii) a gating mechanism which arbitrates between different group-wise and entry-wise pruning. All these cascaded parametrizations are built upon a common latent tensor which is trained end-to-end by minimizing a classification loss and a surrogate tensor rank regularizer. Extensive experiments, conducted on the challenging tasks of action and hand-gesture recognition, show the clear advantage of our proposed semi-structured pruning approach against both structured and unstructured pruning, when taken separately, as well as the related work.

## 1. Introduction

Deep neural networks (DNNs) are nowadays becoming a hotspot in machine learning with increasingly performant models used to approach eclectic pattern recognition tasks (Krizhevsky et al., 2017; He et al., 2016; 2017; Jian et al., 2020; Ronneberger et al., 2015; Jiu & Sahbi, 2017; 2019). These models are also steadily oversized and this makes their deployment on cheap devices, endowed with limited hardware resources, very challenging. In particular, hand-gesture recognition and human computer interaction tasks require fast and lightweight DNNs with high recognition performances. However, DNNs are currently showing some saturated improvement in accuracy while their computational efficiency remains a major issue. Among these DNN models, graph convolutional networks (GCNs) are deemed effective especially on non-euclidean domains including skeleton-data (Zhu et al., 2016b). Two families of GCNs exist in the literature: spectral and spatial. Spectral methods project graph signals from the input to the Fourier domain, achieve convolution, prior to back-project the convolved signals in the input domain (Kipf & Welling, 2016; Levie et al., 2018; Li et al., 2018b; Defferrard et al., 2016; Bruna et al., 2013; Henaff et al., 2015; Chung, 1997; Sahbi, 2021c; Mazari & Sahbi, 2019b). Spatial methods proceed differently by aggregating signals through neighboring nodes, using multi-head attention, prior to achieve convolutions (as inner products) on the resulting node aggregates (Gori et al., 2005; Micheli, 2009; Wu et al., 2020; Hamilton et al., 2017; Knyazev et al., 2019; Sahbi et al., 2011; Sahbi, 2021b;a). Spatial GCNs are known to be more effective compared to spectral ones. Nonetheless, with multi-head attention, spatial GCNs become oversized, computationally overwhelming, and their deployment of cheap devices requires making them lightweight and still effective (Huang et al., 2018a; Sandler et al., 2018; Howard et al., 2017; Tan & Le, 2019; Cai et al., 2019; He et al., 2018a;b; Sahbi, 2021d; 2023).

Several existing works address the issue of lightweight network design, including tensor decomposition (Howard et al., 2019), quantization (Han et al., 2015a), distillation (Hinton et al., 2015; Mirzadeh et al., 2020; Zhang et al., 2018; Ahn et al., 2019; Sahbi & Geman, 2006), neural architecture search (Li et al., 2022) and pruning (LeCun et al., 1989; Has-

---

[1]Sorbonne University, CNRS, LIP6, F-75005, Paris, France. Correspondence to: <hichem.sahbi@sorbonne-universite.fr>.

Accepted to the Workshop on Advancing Neural Network Training at International Conference on Machine Learning (WANT@ICML 2024).

sibi & Stork, 1992; Han et al., 2015b; Sahbi, 2022). Pruning methods are particularly effective, and their general recipe consists in removing connections in order to enable reduced storage and faster inference with a minimal impact on classification performances. One of the mainstream methods is magnitude pruning (MP) (Han et al., 2015a) which removes the smallest weight connections before retraining the pruned networks. Two categories of MP techniques exist in the literature: unstructured (Han et al., 2015b;a) and structured (Li et al., 2016; Liu et al., 2017d). Unstructured methods remove weights individually by ranking them according to the importance of their magnitudes whilst structured approaches zero-out groups of weights (belonging to entire rows, columns, filters or channels) according to the importance of their *aggregated* magnitudes. Unstructured MP results into more flexible, accurate networks, and allows reaching any fine-grained targeted pruning rate but requires dedicated hardware to actually achieve efficient computation. In contrast, structured MP offers a more practical advantage by making the resulting DNNs compatible with standard hardware for efficient computation. However, this comes at the expense of a reduced classification performance and coarse-grained pruning rates. *In order to fully exhibit the potential of these two pruning categories, a more suitable framework should gather the upsides of both structured and unstructured pruning while discarding their downsides to some extent.*

In this paper, we introduce a novel variational MP approach that leverages both structured and unstructured pruning. This method dubbed as *semi-structured* is based on a differentiable cascaded weight parametrization composed of (i) a band-stop mechanism enforcing the prior that the smallest weights should be removed, (ii) a weight-sharing that groups mask entries belonging to the same rows, columns, or channels in a given tensor, and (iii) a gating mechanism which arbitrates between different mask group assignments while maximizing the accuracy of the trained lightweight networks. We also consider a budget loss that allows implementing any targeted fine-grained pruning rate and reducing further the rank of the pruned tensors, resulting into more efficient networks while being closely accurate as shown later in experiments.

## 2. Related work

The following review discusses the related work in variational pruning and skeleton-based recognition, highlighting the limitations that motivate our contributions.

**Variational Pruning.** The general concept behind variational pruning is to learn weights and binary masks that capture the topology of pruned networks. This is achieved by minimizing a global loss that combines a classification error and a regularizer that controls the sparsity (or the cost) of the resulting networks (Liu et al., 2017d; Wen et al., 2016; Louizos et al., 2017). However, these approaches are powerless to implement any given targeted pruning rate without overtrying multiple weighting of the regularizers. Alternative methods explicitly model the network cost using $\ell_0$-based criteria (Louizos et al., 2017; Pan et al., 2016) in order to minimize the discrepancy between the observed and the targeted costs. Existing solutions rely on sampling heuristics or relaxation, which promote sparsity — using different regularizers ($\ell_1/\ell_2$-based, entropy, etc. ) (Gordon et al., 2018; Carreira-Perpiñán & Idelbayev, 2018; Koneru & Vasudevan, 2019; Wiedemann et al., 2019) — but are powerless to implement any given targeted cost exactly and result in overpruning effects leading to disconnected subnetworks. Furthermore, most of the existing solutions, including magnitude pruning (Han et al., 2015a), decouple the training of network topology from weights, making the learning of pruned networks suboptimal. On another hand, existing pruning methods are either structured (Li et al., 2016; Liu et al., 2017d) or unstructured (Han et al., 2015b;a) so their benefit is not fully explored. In contrast to the aforementioned related work, our contribution in this paper seeks to leverage the advantage of both structured and unstructured pruning where the training of masks and weights are coupled on top of shared latent parameters.

**Skeleton-based recognition.** This task has gained increasing interest due to the emergence of sensors like Intel RealSense and Microsoft Kinect. Early methods for hand-gesture and action recognition used RGB (Liu & Yuan, 2018; Hu et al., 2015; Wang & Sahbi, 2013; Yuan et al., 2012; Wang & Sahbi, 2014), depth (Ohn-Bar & Trivedi, 2014; Wang et al., 2015), shape / normals (Oreifej & Liu, 2013; Rahmani & Mian, 2016; Yun et al., 2012; Ji et al., 2014; Li et al., 2015; Zanfir et al., 2013; Sahbi, 2007; Sahbi & Fleuret, 2004), and skeleton-based techniques (Wang et al., 2018). These methods were based on modeling human motions using handcrafted features (Xia et al., 2012; Yang & Tian, 2014), dynamic time warping (Vemulapalli et al., 2014), temporal information (Zhang et al., 2016; Garcia-Hernando & Kim, 2017), and temporal pyramids (Zhu et al., 2016a; Q. De Smedt & Vandeborre, 2016). However, with the resurgence of deep learning, these methods have been quickly overtaken by 2D/3D Convolutional Neural Networks (CNNs) (Feichtenhofer et al., 2016; Nunez et al., 2018a; Mazari & Sahbi, 2019a), Recurrent Neural Networks (RNNs) (Zhu et al., 2016b; Chen et al., 2017; Ke et al., 2017; Liu et al., 2017c; Liu & Yuan, 2018; Wang et al., 2016; Du et al., 2015; Wang & Wang, 2017; Liu et al., 2017b; Nunez et al., 2018b; Shahroudy et al., 2016; Zhang et al., 2017b; Lee et al., 2017; Liu et al., 2016; Maghoumi & LaViola, 2019; Zhang et al., 2017a; Liu et al., 2017a), manifold learn-

ing (Huang & Van Gool, 2017; Huang et al., 2018b; Kacem et al., 2018), attention-based networks (Liu et al., 2021; Weng et al., 2018; Hou et al., 2018; Chen et al., 2019; Song et al., 2017), and GCNs (Huang et al., 2017; Li et al., 2018a; Yan et al., 2018; Wen et al., 2019; Shi et al., 2018; Nguyen et al., 2019; Li et al., 2019; 2020). The recent emergence of GCNs, in particular, has led to their increased use in skeleton-based recognition (Li et al., 2018b). These models capture spatial and temporal attention among skeleton joints with better interpretability. However, when tasks involve relatively large input graphs, GCNs (particularly with multi-head attention) become computationally inefficient and require lightweight design techniques. In this paper, we design efficient GCNs that make skeleton-based recognition highly efficient while also being effective.

## 3. A Glimpse on Graph Convolutional Networks

Let $\mathcal{S} = \{\mathcal{G}_i = (\mathcal{V}_i, \mathcal{E}_i)\}_i$ denote a collection of graphs with $\mathcal{V}_i, \mathcal{E}_i$ being respectively the nodes and the edges of $\mathcal{G}_i$. Each graph $\mathcal{G}_i$ (denoted for short as $\mathcal{G} = (\mathcal{V}, \mathcal{E})$) is endowed with a signal $\{\phi(u) \in \mathbb{R}^s : u \in \mathcal{V}\}$ and associated with an adjacency matrix $\mathbf{A}$. GCNs aim at learning a set of $C$ filters $\mathcal{F}$ that define convolution on $n$ nodes of $\mathcal{G}$ (with $n = |\mathcal{V}|$) as $(\mathcal{G} \star \mathcal{F})_\mathcal{V} = f(\mathbf{A} \, \mathbf{U}^\top \, \mathbf{W})$, here $^\top$ stands for transpose, $\mathbf{U} \in \mathbb{R}^{s \times n}$ is the graph signal, $\mathbf{W} \in \mathbb{R}^{s \times C}$ is the matrix of convolutional parameters corresponding to the $C$ filters and $f(.)$ is a nonlinear activation applied entry-wise. In $(\mathcal{G} \star \mathcal{F})_\mathcal{V}$, the input signal $\mathbf{U}$ is projected using $\mathbf{A}$ and this provides for each node $u$, the aggregate set of its neighbors. Entries of $\mathbf{A}$ could be handcrafted or learned so $(\mathcal{G} \star \mathcal{F})_\mathcal{V}$ corresponds to a convolutional block with two layers; the first one aggregates signals in $\mathcal{N}(\mathcal{V})$ (sets of node neighbors) by multiplying $\mathbf{U}$ with $\mathbf{A}$ while the second layer achieves convolution by multiplying the resulting aggregates with the $C$ filters in $\mathbf{W}$. Learning multiple adjacency (also referred to as attention) matrices (denoted as $\{\mathbf{A}^k\}_{k=1}^K$) allows us to capture different contexts and graph topologies when achieving aggregation and convolution. With multiple matrices $\{\mathbf{A}^k\}_k$ (and associated convolutional filter parameters $\{\mathbf{W}^k\}_k$), $(\mathcal{G} \star \mathcal{F})_\mathcal{V}$ is updated as $f\left(\sum_{k=1}^K \mathbf{A}^k \mathbf{U}^\top \mathbf{W}^k\right)$. Stacking aggregation and convolutional layers, with multiple matrices $\{\mathbf{A}^k\}_k$, makes GCNs accurate but heavy. We propose, in what follows, a method that makes our networks lightweight and still effective.

## 4. Proposed Method: Semi-Structured Magnitude Pruning

In what follows, we formally subsume a given GCN as a multi-layered neural network $g_\theta$ whose weights are defined as $\theta = \{\mathbf{W}^1, \dots, \mathbf{W}^L\}$, being $L$ its depth, $\mathbf{W}^\ell \in$ $\mathbb{R}^{d_{\ell-1} \times d_\ell}$ its $\ell^{\text{th}}$ layer weight tensor, and $d_\ell$ the dimension of $\ell$. The output of a given layer $\ell$ is defined as $\phi^\ell = f_\ell(\mathbf{W}^{\ell^\top} \phi^{\ell-1})$, $\ell \in \{2, \dots, L\}$, with $f_\ell$ an activation function; without a loss of generality, we omit the bias in the definition of $\phi^\ell$.

Pruning consists in zeroing-out a subset of weights in $\theta$ by multiplying $\mathbf{W}^\ell$ with a binary mask $\mathbf{M}^\ell \in \{0, 1\}^{d_{\ell-1} \times d_\ell}$. The binary entries of $\mathbf{M}^\ell$ are set depending on whether the underlying layer connections are pruned, so $\phi^\ell = f_\ell((\mathbf{M}^\ell \odot \mathbf{W}^\ell)^\top \phi^{\ell-1})$, here $\odot$ stands for the element-wise matrix product. In our definition of semi-structured pruning, entries of the tensor $\{\mathbf{M}^\ell\}_\ell$ are set depending on the prominence and also on how the underlying connections in $g_\theta$ are grouped; pruning that removes the entire connections individually (resp. jointly) is referred to as *unstructured* (resp. *structured*) whereas pruning that removes some connections independently and others jointly is dubbed as *semi-structured*. However, such pruning (with $\{\mathbf{M}^\ell\}_\ell$) suffers from several drawbacks. In the one hand, optimizing the discrete set of variables $\{\mathbf{M}^\ell\}_\ell$ is deemed highly combinatorial and intractable especially on large networks. In the other hand, the total number of parameters $\{\mathbf{M}^\ell\}_\ell$, $\{\mathbf{W}^\ell\}_\ell$ is twice the number of connections in $g_\theta$ and this increases training complexity and may also lead to overfitting.

### 4.1. Semi-Structured Weight Parametrization

In order to overcome the aforementioned issues, we consider an alternative *parametrization* that allows finding both the topology of the pruned networks together with their weights, without doubling the size of the training parameters, while making magnitude pruning semi-structured and learning still effective. This parametrization corresponds to the Hadamard product involving a weight tensor and a *cascaded* function applied to the same tensor as

$$\mathbf{W}^\ell = \hat{\mathbf{W}}^\ell \odot \left[\psi_3 \circ \psi_2 \circ \psi_1(\hat{\mathbf{W}}^\ell)\right], \qquad (1)$$

being $\hat{\mathbf{W}}^\ell$ a latent tensor and $\psi(\hat{\mathbf{W}}^\ell)$ (with $\psi = \psi_3 \circ \psi_2 \circ \psi_1$) a continuous relaxation of $\mathbf{M}^\ell$ which enforces the prior that (i) smallest weights $\hat{\mathbf{W}}^\ell$ should be removed from the network, (ii) the underlying mask entries $\psi(\hat{\mathbf{W}}^\ell)$ are shared (across tensor rows, columns, channels, etc.) when pruning is structured, and (iii) any given mask entry in $\psi(\hat{\mathbf{W}}^\ell)$ is either unstructurally or structurally pruned. In what follows, we detail the different parametrizations used to define $\psi(\hat{\mathbf{W}}^\ell)$; unless explicitly mentioned, we omit $\ell$ in the definition of $\hat{\mathbf{W}}^\ell$ and we rewrite it simply as $\hat{\mathbf{W}}$.

**Band-stop Parametrization** ($\psi_1$). This parametrization $\psi_1$ is entry-wise applied to the tensor $\hat{\mathbf{W}}$ and enforces the prior that smallest weights should be removed from the network. In order to achieve this goal, $\psi_1$ must be (i) bounded in $[0, 1]$,

(ii) differentiable, (iii) symmetric, and (iv) $\psi_1(\omega) \rightsquigarrow 1$ when $|\omega|$ is sufficiently large and $\psi_1(\omega) \rightsquigarrow 0$ otherwise. The first and the fourth properties ensure that the parametrization is neither acting as a scaling factor greater than one nor changing the sign of the latent weight, and also acts as the identity for sufficiently large weights, and as a contraction factor for small ones. The second property is necessary to ensure that $\psi_1$ has computable gradient while the third condition guarantees that only the magnitudes of the latent weights matter. A choice, used in practice, that satisfies these four conditions is

$$\psi_1(\omega) = 2\big(1 + \exp(-\sigma\omega^2)\big)^{-1} - 1, \qquad (2)$$

being $\sigma$ a scaling factor that controls the crispness (binarization) of mask entries in $\psi_1(\hat{\mathbf{W}})$. According to Eq. 2, $\sigma$ controls the smoothness of $\psi_1$ around the support of the latent weights. This allows implementing an annealed (soft) thresholding function that cuts-off all the connections in a smooth and differentiable manner as training of the latent parameters evolves. The asymptotic behavior of $\psi_1$ — that allows selecting the topology of the pruned subnetworks — is obtained as training reaches the latest epochs, and this makes mask entries, in $\psi_1(\hat{\mathbf{W}})$, crisp and (almost) binary. This mask $\psi_1(\hat{\mathbf{W}})$ (rewritten for short as $\psi_1$) is used as input to the subsequent parameterizations $\psi_2$ and $\psi_3$ as shown below.

**Weight-sharing Parametrization ($\psi_2$).** This parametrization $\psi_2$ implements semi-structured pruning by *tying* mask entries belonging to the same rows, columns or channels in the tensor $\psi_1$. More precisely, each mask entry in $\psi_2(\psi_1)$ will either be (i) entry-wise evaluated (dependent only on its underlying weight), or (ii) shared through multiple latent weights belonging to the same row, column or channel of $\psi_1$ resulting into the following multi-head parametrization (see Fig. 1)

$$\psi_2(\psi_1) = \begin{cases} \psi_2^u(\psi_1) \ = \psi_1 \\ \qquad\qquad \text{unstructured (entry-wise)} \\ \psi_2^r(\psi_1) \ = \mathbf{vec}^{-1}(\mathbf{P}_r\,\mathbf{vec}(\psi_1)) \\ \qquad\qquad \text{structured (row-wise)} \\ \psi_2^c(\psi_1) \ = \mathbf{vec}^{-1}(\mathbf{vec}(\psi_1)^\top\,\mathbf{P}_c) \\ \qquad\qquad \text{structured (column-wise)} \\ \psi_2^b(\psi_1) \ = \mathbf{vec}^{-1}(\mathbf{P}_r\mathbf{P}_c^\top\,\mathbf{vec}(\psi_1)) \\ \qquad\qquad \text{structured (block/channel-wise),} \end{cases} \qquad (3)$$

here $\mathbf{vec}$ (resp. $\mathbf{vec}^{-1}$) reshapes a matrix into a vector (resp. vice-versa), and $\mathbf{P}_r \in \{0,1\}^{(d_{\ell-1} \times d_\ell)^2}$, $\mathbf{P}_c \in \{0,1\}^{(d_{\ell-1} \times d_\ell)^2}$ are two (layer-wise fixed) adjacency matrices that model the neighborhood system across respectively the rows and the columns of $\psi_1$ (in other words, $[\mathbf{P}_r]_{ij,pq} \neq 0$ iff the two connections $(i,j)$, $(p,q)$, in a

given layer, share the same neuron, i.e., $i = p$) whilst the product $\mathbf{P}_r\mathbf{P}_c^\top \in \{0,1\}^{(d_{\ell-1} \times d_\ell)^2}$ models this neighborhood through blocks/channels of $\psi_1$. When composed (with $\psi_1$), the mask $\psi_2$ inherits all the aforementioned fourth properties: mask entries in $\psi_2(\psi_1)$ remain bounded in $[0,1]$, differentiable, symmetric, and close to 1 when entries of the latent tensor $\hat{\mathbf{W}}$ (i.e., inputs of $\psi_1$) are sufficiently large and 0 otherwise.

**Gating Parametrization ($\psi_3$).** As each connection in $g_\theta$ is endowed with a multi-head parametrization $\psi_2$, we define $\psi_3$ as a gating mechanism that selects only one of them. More precisely, each mask entry can either be (i) entry-wise pruned, i.e., untied, or (ii) tied to its row, column or block/channel. Again with $\psi_3$, the composed parametrization $\psi_3(\psi_2)$ is bounded in $[0,1]$, differentiable, symmetric and reaches 1 if the entries of the latent tensor $\hat{\mathbf{W}}$ are sufficiently large, and 0 otherwise. Formally, the gating mechanism $\psi_3$ is defined as

$$\psi_3(\psi_2) = \underbrace{\psi_2^b}_{\text{block-wise}} + \underbrace{(\bar{\psi}_2^b) \odot \psi_2^c}_{\text{column-wise}} + \underbrace{(\bar{\psi}_2^b \odot \bar{\psi}_2^c) \odot \psi_2^r}_{\text{row-wise}} \\ + \underbrace{(\bar{\psi}_2^b \odot \bar{\psi}_2^c \odot \bar{\psi}_2^r) \odot \psi_2^u}_{\text{entry-wise}}, \qquad (4)$$

being $\bar{\psi}_2^b = \mathbf{U} - \psi_2^b$, and $\mathbf{U}$ a tensor of ones with the same dimensions as $\psi_2^b$ (and $\bar{\psi}_2^r$, $\bar{\psi}_2^c$, $\bar{\psi}_2^u$ are similarly defined). It is easy to see that when entries in $\psi_1$ (and hence $\psi_2$) are crisp, at most one of these four terms is activated (i.e., equal to one) for each connection in $g_\theta$. From Eq. 4, block-wise pruning has the highest priority, followed by column-wise, row-wise and then entry-wise pruning respectively. This priority allows designing highly efficient lightweight networks with a coarse-granularity budget implementation for block/column/row-wise (structured) pruning, while entry-wise (unstructured) pruning is less computationally efficient but allows reaching any targeted budget with a finer granularity, and thereby with a better accuracy. Note that this parametrization acts as a weight regularizer which not only improves the lightweightness of the pruned networks but also their generalization performances (as shown later in experiments). Note also that $\psi_1$ and $\psi_2$ are commutable in the cascaded parametrization $\psi = \psi_3 \circ \psi_2 \circ \psi_1$ but $\psi_3$ should be applied at the end of the cascade.

### 4.2. Budget-Aware Variational Pruning

By considering Eq. 1, we define our semi-structured pruning loss as

$$\mathcal{L}_e\big(\{\psi_3 \circ \psi_2 \circ \psi_1(\hat{\mathbf{W}}^\ell) \odot \hat{\mathbf{W}}^\ell\}_\ell\big) \\ + \lambda\bigg(\sum_{\ell=1}^{L-1} 1_{d_\ell}^\top\,[\psi_3 \circ \psi_2 \circ \psi_1(\hat{\mathbf{W}}^\ell)]\,1_{d_{\ell+1}} - c\bigg)^2, \qquad (5)$$

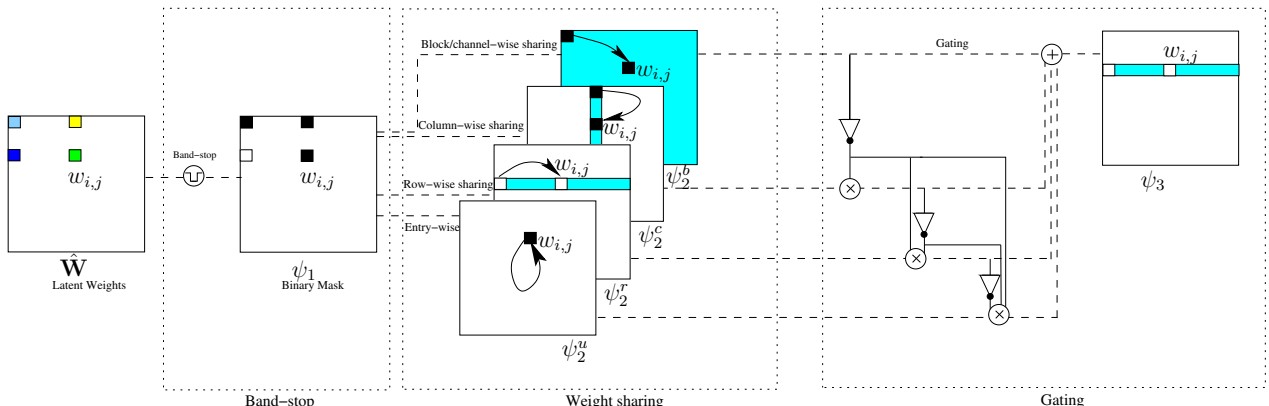

Figure 1: This figure shows the three stages of the cascaded parametrization including (i) band-stop, (ii) weight-sharing and (iii) gating. Cyan stands for shared connections, and the triangle for the "not gate" operator. For ease of visualization, only 4 connections are shown during the whole evaluation of the parameterization, and only the outcome (1 or 0) of $w_{i,j}$ is shown in the final mask tensor.

being $1_{d_\ell}$ a vector of $d_\ell$ ones, and the left-hand side term is the cross entropy loss that measures the discrepancy between predicted and ground-truth labels. The right-hand side term is a budget loss that allows reaching any targeted pruning cost $c$. Nonetheless, it's worth noticing that actual efficiency is not only related to the pruning rate but also to the actual dimensionality of the tensors. In order to take full advantage of the semi-structured setting of our method, we complement the aforementioned budget function with another one that minimizes the rank of the pruned tensors $\{\psi_3 \circ \psi_2 \circ \psi_1(\hat{\mathbf{W}}^\ell)\}_\ell$. However, as the rank is not differentiable, we consider a surrogate function (as an upper bound) of the rank[1]. Hence, Eq. 5 becomes

$$
\begin{aligned}
&\mathcal{L}_e\big(\psi_3 \circ \psi_2 \circ \psi_1(\hat{\mathbf{W}}^\ell) \odot \{\hat{\mathbf{W}}^\ell\}_\ell\big) \\
&+\lambda\bigg(\sum_{\ell=1}^{L-1} 1_{d_\ell}^\top [\psi_3 \circ \psi_2 \circ \psi_1(\hat{\mathbf{W}}^\ell)]\, 1_{d_{\ell+1}} - c\bigg)^2 \\
&+\beta \sum_{\ell=1}^{L-1} r[(\psi_3 \circ \psi_2 \circ \psi_1(\hat{\mathbf{W}}^\ell)],
\end{aligned}
\tag{6}
$$

being $r[\mathbf{W}]$ a surrogate differentiable rank function set in practice to

$$
r[\mathbf{W}] = \begin{aligned}[t] &[1_{d_{\ell+1}}^\top - \exp(-\gamma 1_{d_\ell}^\top \mathbf{W})]1_{d_{\ell+1}} \\ &+1_{d_\ell}^\top [1_{d_\ell} - \exp(-\gamma \mathbf{W} 1_{d_{\ell+1}})], \end{aligned}
\tag{7}
$$

being $\gamma$ an annealed temperature and $\exp(.)$ is entry-wise applied. Eq. 7 seeks to minimize the number of non-null rows/columns in a given tensor $\mathbf{W}$, and this allows achieving higher speedup compared to when *only* the budget loss is minimized (see experiments). In Eq. 6, $\beta$ controls the

---

[1]This function is an upper bound of the rank as it corresponds to the sum of the number of non-zero rows and non-zero columns.

"structureness" of pruning; large $\beta$ favors stringent tensors first through blocks, columns and then through rows, while smaller $\beta$ leads to *mixed* structured and unstructured pruning. Once the above loss optimized, actual rank minimization requires reordering dimensions layer-wise in order to fully benefit from compact tensors and eliminate fragmentation; this is achievable as only outward connections, from unpruned neurons in each layer, are actually pruned during optimization.

### 4.3. Optimization

Let $\mathcal{L}$ denote the global loss in Eq. 6, the update of $\{\hat{\mathbf{W}}^\ell\}_\ell$ is achieved using the gradient of $\mathcal{L}$ obtained by propagating the gradients through $g_\theta$. More precisely, considering the parametrization in Eq. 1, the gradient of the global loss w.r.t. $\hat{\mathbf{W}}^\ell$ is obtained as

$$
\frac{\partial \mathcal{L}}{\partial \hat{\mathbf{W}}^\ell} = \frac{\partial \mathcal{L}}{\partial \psi(\hat{\mathbf{W}}^\ell)} \frac{\partial \psi(\hat{\mathbf{W}}^\ell)}{\partial \psi_2 \circ \psi_1(\hat{\mathbf{W}}^\ell)} \frac{\partial \psi_2 \circ \psi_1(\hat{\mathbf{W}}^\ell)}{\partial \psi_1(\hat{\mathbf{W}}^\ell)} \frac{\partial \psi_1(\hat{\mathbf{W}}^\ell)}{\partial \hat{\mathbf{W}}^\ell},
\tag{8}
$$

here the original gradient $\partial\mathcal{L}/\partial\psi(\hat{\mathbf{W}}^\ell)$ is obtained from layer-wise backpropagation, and $\frac{\partial\mathcal{L}}{\partial\hat{\mathbf{W}}^\ell}$ is obtained by multiplying the original gradient by the three rightmost Jacobians (whose matrix forms are shown in Table 1).

In the above objective function, $\beta = 0.1$ and $\lambda$ is overestimated (to 1000 in practice) in order to make Eq. 6 focusing on the implementation of the budget. As training reaches its final epochs, the budget loss reaches its minimum and the gradient of the global objective function will be dominated by the gradient of $\mathcal{L}_e$ (and to some extent by the gradient of the surrogate rank function); this allows improving both classification performances and efficiency as shown subsequently.

| Entry-wise | Row-wise | Column-wise | Block-wise |
|---|---|---|---|
| $[\mathbf{J}_1]_{ij,pq} = 1_{\{ij=pq\}}\psi_1'(\hat{\mathbf{W}}_{pq})$ | NA | NA | NA |
| $[\mathbf{J}_2^u]_{ij,pq} = 1_{\{ij=pq\}}$ | $[\mathbf{J}_2^r]_{ij,pq} = [\mathbf{P}_r]_{ij,pq}$ | $[\mathbf{J}_2^c]_{ij,pq} = [\mathbf{P}_c']_{ij,pq}$ | $[\mathbf{J}_2^b]_{ij,pq} = [\mathbf{P}_r\mathbf{P}_c']_{ij,pq}$ |
| $[\mathbf{J}_3^u]_{ij,pq} = 1_{\{ij=pq\}}$ | $[\mathbf{J}_3^r]_{ij,pq} = 1_{\{ij=pq\}}$ | $[\mathbf{J}_3^c]_{ij,pq} = 1_{\{ij=pq\}}$ | $[\mathbf{J}_3^b]_{ij,pq} = 1_{\{ij=pq\}}$ |
| $\times [\bar{\psi}_2^b \odot \bar{\psi}_2^c \odot \bar{\psi}_2^r]_{pq}$ | $\times [\bar{\psi}_2^b \odot \bar{\psi}_2^c \odot \bar{\psi}_2^u]_{pq}$ | $\times [\bar{\psi}_2^b \odot \bar{\psi}_2^r \odot \bar{\psi}_2^u]_{pq}$ | $\times [\bar{\psi}_2^c \odot \bar{\psi}_2^r \odot \bar{\psi}_2^u]_{pq}$ |

Table 1: Jacobians of different parametrizations w.r.t. different settings; here $[\mathbf{J}_1]_{ij,pq} = [\partial\psi_1/\partial\hat{\mathbf{W}}]_{ij,pq}$, $[\mathbf{J}_2^{\mathbf{x}}]_{ij,pq} = [\partial\psi_2^{\mathbf{x}}/\partial\psi_1]_{ij,pq}$ and $[\mathbf{J}_3^{\mathbf{x}}]_{ij,pq} = [\partial\psi_3/\partial\psi_2^{\mathbf{x}}]_{ij,pq}$ with $\mathbf{x} \in \{u, r, c, b\}$; here $u$, $r$, $c$ and $b$ stand for entry-wise, row-wise, column-wise and block-wise respectively. It is easy to see that all these Jacobians are extremely sparse and their evaluation is highly efficient. In this table, NA stands for not applicable as the Jacobian of the parametrization $\psi_1$ is necessarily entry-wise.

# 5. Experiments

This section assesses baseline and pruned GCNs' performance in skeleton-based recognition using SBU Interaction (Yun et al., 2012) and the First Person Hand Action (FPHA) (Garcia-Hernando et al., 2018) datasets, comparing our lightweight GCNs against related pruning techniques. SBU is an interaction dataset acquired using the Microsoft Kinect sensor, it contains 282 moving skeleton sequences performed by two interacting individuals and belonging to 8 categories. Each pair of interacting individuals corresponds to two 15 joint skeletons, each one encoded with a sequence of its 3D coordinates across video frames. The evaluation protocol follows the train-test split as in the original dataset release (Yun et al., 2012). The FPHA dataset includes 1175 skeletons belonging to 45 action categories performed by 6 different individuals in 3 scenarios. Action categories are highly variable, including various styles, speed, scale, and viewpoint. Each skeleton includes 21 hand joints, each one again encoded with a sequence of its 3D coordinates across video frames. The performances of different methods are evaluated using the 1:1 setting proposed in (Garcia-Hernando et al., 2018) with 600 action sequences for training and 575 for testing. The average accuracy over all classes of actions is reported in all experiments.

**Input graphs.** Let's consider a sequence of skeletons $\{S^t\}_t$ with $S^t = \{\hat{p}_1^t, \ldots, \hat{p}_n^t\}$ being the 3D skeleton coordinates at frame $t$, and $\{\hat{p}_j^t\}_t$ a joint trajectory through successive frames. We define an input graph $\mathcal{G} = (\mathcal{V}, \mathcal{E})$ as a finite collection of trajectories, with each node $v_j \in \mathcal{V}$ in $\mathcal{G}$ being a trajectory $\{\hat{p}_j^t\}_t$, and an edge $(v_j, v_i) \in \mathcal{E}$ exists between two nodes if the underlying trajectories are spatially neighbors. Each trajectory is processed using *temporal chunking*, which splits the total duration of a sequence into $M$ evenly-sized temporal chunks (with $M = 4$ in practice). Then, joint coordinates $\{\hat{p}_j^t\}_t$ of the trajectory are assigned to these chunks, based on their time stamps. The averages of these chunks are concatenated in order to create the raw description of the trajectory (denoted as $\phi(v_j) \in \mathbb{R}^s$ with $s = 3 \times M$). This process preserves the temporal structure

| Method | Accuracy (%) |
|---|---|
| Raw Position (Yun et al., 2012) | 49.7 |
| Joint feature (Ji et al., 2014) | 86.9 |
| CHARM (Li et al., 2015) | 86.9 |
| H-RNN (Du et al., 2015) | 80.4 |
| ST-LSTM (Liu et al., 2016) | 88.6 |
| Co-occurrence-LSTM (Zhu et al., 2016b) | 90.4 |
| STA-LSTM (Song et al., 2017) | 91.5 |
| ST-LSTM + Trust Gate (Liu et al., 2016) | 93.3 |
| VA-LSTM (Zhang et al., 2017a) | 97.6 |
| GCA-LSTM (Liu et al., 2017a) | 94.9 |
| Riemannian manifold. traj (Kacem et al., 2018) | 93.7 |
| DeepGRU (Maghoumi & LaViola, 2019) | 95.7 |
| RHCN + ACSC + STUFE (Li et al., 2020) | 98.7 |
| Our baseline (unpruned) GCN | 98.4 |

Table 2: Comparison of our baseline GCN against related work on the SBU database.

| Method | Color | Depth | Pose | Accuracy (%) |
|---|---|---|---|---|
| 2-stream-color (Feichtenhofer et al., 2016) | ✓ | ✗ | ✗ | 61.56 |
| 2-stream-flow (Feichtenhofer et al., 2016) | ✓ | ✗ | ✗ | 69.91 |
| 2-stream-all (Feichtenhofer et al., 2016) | ✓ | ✗ | ✗ | 75.30 |
| HOG2-dep (Ohn-Bar & Trivedi, 2014) | ✗ | ✓ | ✗ | 59.83 |
| HOG2-dep+pose (Ohn-Bar & Trivedi, 2014) | ✗ | ✓ | ✓ | 66.78 |
| HON4D (Oreifej & Liu, 2013) | ✗ | ✓ | ✗ | 70.61 |
| Novel View (Rahmani & Mian, 2016) | ✗ | ✓ | ✗ | 69.21 |
| 1-layer LSTM (Zhu et al., 2016b) | ✗ | ✗ | ✓ | 78.73 |
| 2-layer LSTM (Zhu et al., 2016b) | ✗ | ✗ | ✓ | 80.14 |
| Moving Pose (Zanfir et al., 2013) | ✗ | ✗ | ✓ | 56.34 |
| Lie Group (Vemulapalli et al., 2014) | ✗ | ✗ | ✓ | 82.69 |
| HBRNN (Du et al., 2015) | ✗ | ✗ | ✓ | 77.40 |
| Gram Matrix (Zhang et al., 2016) | ✗ | ✗ | ✓ | 85.39 |
| TF (Garcia-Hernando & Kim, 2017) | ✗ | ✗ | ✓ | 80.69 |
| JOULE-color (Hu et al., 2015) | ✓ | ✗ | ✗ | 66.78 |
| JOULE-depth (Hu et al., 2015) | ✗ | ✓ | ✗ | 60.17 |
| JOULE-pose (Hu et al., 2015) | ✗ | ✗ | ✓ | 74.60 |
| JOULE-all (Hu et al., 2015) | ✓ | ✓ | ✓ | 78.78 |
| Huang et al. (Huang & Van Gool, 2017) | ✗ | ✗ | ✓ | 84.35 |
| Huang et al. (Huang et al., 2018b) | ✗ | ✗ | ✓ | 77.57 |
| HAN (Liu et al., 2021) | ✗ | ✗ | ✓ | 85.74 |
| Our baseline (unpruned) GCN | ✗ | ✗ | ✓ | 86.43 |

Table 3: Comparison of our baseline GCN against related work on the FPHA database.

of trajectories while being frame-rate and duration agnostic.

**Implementation details & baseline GCNs.** All the GCNs have been trained using the Adam optimizer for $2,700$ epochs with a batch size of $200$ for SBU and $600$ for FPHA, a momentum of $0.9$, and a global learning rate (denoted as $\nu(t)$) inversely proportional to the speed of change of the loss used to train the networks; with $\nu(t)$ decreasing as $\nu(t) \leftarrow \nu(t-1) \times 0.99$ (resp. increasing as $\nu(t) \leftarrow \nu(t-1)/0.99$) when the speed of change of the loss in Eq. 6 increases (resp. decreases). Experiments were run on a GeForce GTX 1070 GPU device with 8 GB memory, without dropout or data augmentation. The baseline GCN architecture for SBU includes an attention layer of 8 heads, a convolutional layer of 16 filters, a dense fully connected layer, and a softmax layer. The baseline GCN architecture for FPHA is heavier and includes 16 heads, a convolutional layer of 32 filters, a dense fully connected layer, and a softmax layer. Both the baseline GCN architectures, on the SBU and the FPHA benchmarks, are accurate (see tables. 2 and 3), and our goal is to make them lightweight while maintaining their high accuracy.

| Pruning rates | Accuracy (%) | SpeedUp | Observation |
|---|---|---|---|
| 0% | 98.40 | none | Baseline (unpruned) GCN |
| 70% | 93.84 | none | Band-stop Weight Param. |
| 90% | 87.69 | 426× | Structured |
| | 89.23 | 487× | Structured (+ rank optimization) |
| | 93.84 | none | Unstructured |
| | 93.84 | 16× | Unstructured (+ rank optimization) |
| | 90.76 | 40× | Semi-structured |
| | 89.23 | 52× | Semi-structured (+ rank optimization) |
| 95% | 87.69 | 678× | Structured |
| | 87.69 | 787× | Structured (+ rank optimization) |
| | 92.30 | none | Unstructured |
| | 92.30 | 16× | Unstructured (+ rank optimization) |
| | 92.30 | 109× | Semi-structured |
| | 93.84 | 106× | Semi-structured (+ rank optimization) |
| 98% | 81.53 | 797× | Structured |
| | 81.53 | 2195× | Structured (+ rank optimization) |
| | 89.23 | none | Unstructured |
| | 89.23 | 106× | Unstructured (+ rank optimization) |
| | 83.07 | 135× | Semi-structured |
| | 86.15 | 607× | Semi-structured (+ rank optimization) |
| Comparative (regularization-based) pruning | | | |
| 98% | 55.38 | none | MP+$\ell_0$-reg. |
| | 73.84 | none | MP+$\ell_1$-reg. |
| | 61.53 | none | MP+Entropy-reg. |
| | 75.38 | none | MP+Cost-aware-reg. |

Table 4: This table shows detailed performances and ablation study on SBU for different pruning rates. "none" stands for no-actual speedup is observed as the underlying tensors/architecture remain shaped identically to the unpruned network (despite having pruned connections). For structured, unstructured and semi-structured settings, when "rank optimization" is not used, only pruning rate is considered in the loss together with cross entropy. When "rank optimization" is used, all the three terms are combined in the loss.

| Pruning rates | Accuracy (%) | SpeedUp | Observation |
|---|---|---|---|
| 0% | 86.43 | none | Baseline (unpruned) GCN |
| 50% | 85.56 | none | Band-stop Weight Param. |
| 90% | 68.00 | 274× | Structured |
| | 71.30 | 547× | Structured (+ rank optimization) |
| | 83.82 | none | Unstructured |
| | 84.17 | 16× | Unstructured (+ rank optimization) |
| | 78.60 | 33× | Semi-structured |
| | 80.52 | 38× | Semi-structured (+ rank optimization) |
| 95% | 56.69 | 759× | Structured |
| | 62.60 | 931× | Structured (+ rank optimization) |
| | 78.78 | none | Unstructured |
| | 80.17 | 29× | Unstructured (+ rank optimization) |
| | 72.17 | 197× | Semi-structured |
| | 74.60 | 214× | Semi-structured (+ rank optimization) |
| 98% | 47.47 | 1479× | Structured |
| | 49.04 | 1399× | Structured (+ rank optimization) |
| | 78.08 | none | Unstructured |
| | 77.56 | 126× | Unstructured (+ rank optimization) |
| | 75.13 | 33× | Semi-structured |
| | 73.91 | 278× | Semi-structured (+ rank optimization) |
| Comparative (regularization-based) pruning | | | |
| 98% | 64.69 | none | MP+$\ell_0$-reg. |
| | 70.78 | none | MP+$\ell_1$-reg. |
| | 67.47 | none | MP+Entropy-reg. |
| | 69.91 | none | MP+Cost-aware-reg. |

Table 5: This table shows detailed performances and ablation study on FPHA for different pruning rates. "none" stands for no-actual speedup is observed as the underlying tensors/architecture remain shaped identically to the unpruned network (despite having pruned connections).

**Performances, Comparison & Ablation.** Tables 4-5 show a comparison and an ablation study of our method both on the SBU and the FPHA datasets. First, according to the observed results, when only the cross entropy loss is used without budget (i.e., $\lambda = \beta = 0$ in Eq. 6), performances are close to the initial heavy GCNs (particularly on FPHA), with less parameters[2] as this produces a regularization effect similar to (Wan et al., 2013). Then, when pruning is structured, the accuracy is relatively low but the speedup is important particularly for high pruning regimes. When pruning is unstructured, the accuracy reaches its highest value, but no actual speedup is observed as the architecture of the pruned networks remains unchanged (i.e., not compact). When pruning is semi-structured, we observe the best trade-off between accuracy and speedup; in other words, *coarsely* pruned parts of the network (related to entire block/column/row connections) lead to high speedup and efficient computation, whereas *finely* pruned parts (related to individual connections) lead to a better accuracy with a contained marginal impact on computation, so speedup is still globally observed with a significant amount. Extra comparison of our method against other regularizers shows a substantial gain. Indeed, our method is compared against different variational pruning with regularizers plugged in Eq. 6 (instead of our proposed budget and rank

---

[2]Pruning rate does not exceed 70% and no control on this rate is achievable when $\lambda = 0$.

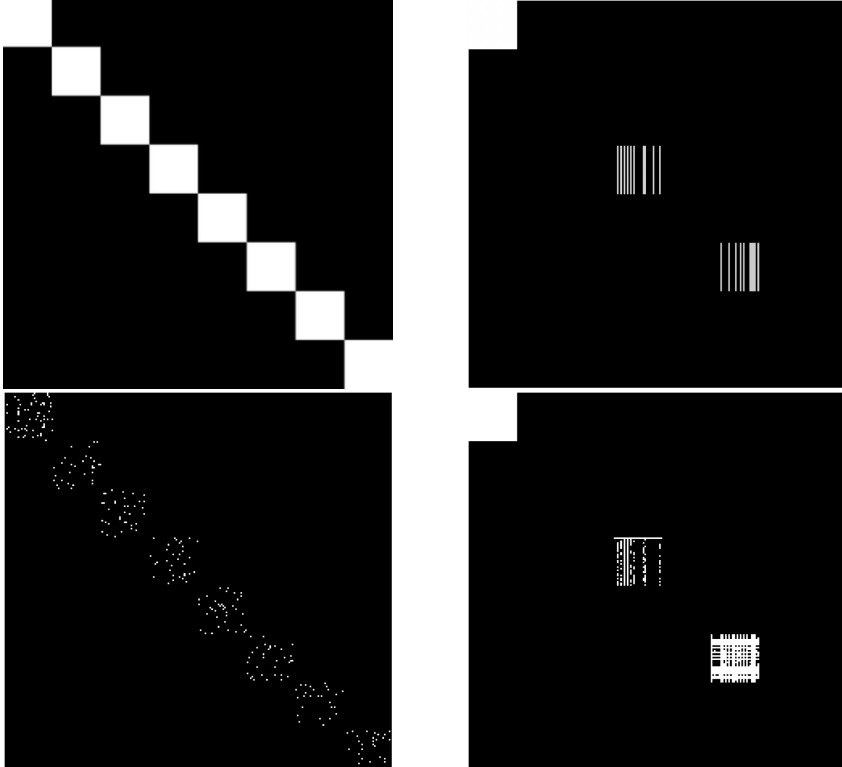

Figure 2: This figure shows a crop of the mask tensor obtained after the gating parametrization when trained on the FPHA dataset. Top-left corresponds to the original mask (without pruning) while the others correspond to masks obtained with structured, unstructured and semi-structured pruning respectively. In all these masks, each diagonal block corresponds to a channel. Better to zoom the PDF.

regularizers), namely $\ell_0$ (Louizos et al., 2017), $\ell_1$ (Koneru & Vasudevan, 2019), entropy (Wiedemann et al., 2019) and $\ell_2$-based cost (Lemaire et al., 2019), all without our parametrization. From the observed results, the impact of our method is substantial for different settings and for equivalent pruning rate (namely 98%). Note that when alternative regularizers are used, multiple settings (trials) of the underlying mixing hyperparameters (in Eq. 6) are considered prior to reach the targeted pruning rate, and this makes the whole training and pruning process overwhelming. While cost-aware regularization makes training more affordable, its downside resides in the observed collapse of trained masks; this is a well known effect that affects performances at high pruning rates. Finally, Fig.2 shows examples of obtained mask tensors taken from the second (attention) layer of the pruned GCN. For semi-structured pruning, we observe a compact tensor layer with some individually pruned connections whereas structured and unstructured pruning — when applied separately — either produce *compact* or *spread* tensors, with a negative impact on respectively *accuracy* or *speed*. In sum, semi-structured pruning gathers the advantages of *both* while discarding their inconveniences.

## 6. Conclusion

This paper introduces a novel magnitude pruning approach that combines *both* the strengths of structured and unstructured pruning methods while discarding their drawbacks. The proposed method, dubbed as *semi-structured*, is based on a novel cascaded weight parametrization including band-stop, weight-sharing, and gating mechanisms. Our pruning method also relies on a budget loss that allows implementing fine-grained targeted pruning rates while also reducing the rank of the pruned tensors resulting in more efficient and still effective networks. Extensive experiments, conducted on the challenging task of skeleton-based recognition, corroborate all these findings.

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
