# OpenReview forum: "Coarse-to-Fine Semi-Structured Pruning of Graph Convolutional Networks for Skeleton-based Recognition"
_ICML.cc/2024/Workshop/WANT — WANT@ICML 2024 Poster_

### Official Review · Reviewer_jZvT · 2024-06-11
**A novel semi-structured technique, however, the experiment section needs to be revised.**

**Confidence:** 3

**Summary:**

Overall rating: Borderline accept

**Strengths:**

This paper proposes a GCN-based skeleton recognition model with Coarse-to-Fine Semi-Structured Pruning optimization to speed up the training and inference process.
To extend traditional MP approaches that leverage both structured and unstructured pruning.
The proposed solution exploits a semi-structured pruning technique that applies
a band-stop mechanism, a weight-sharing, and a gating mechanism.

S1: The paper is well organized and includes comprehensive related works
to introduce the existing techniques for the problem to the audience.

S2:  The semi-structured technique is novel and can speed up the process
According to the experiments.

**Weaknesses:**

W1: The introduction section may need to point out the advantages of
GCN-based solution for the skeleton recognition problem compared with traditional CNN-based solution.
Skeleton pruning is a well-studied problem also for traditional CNN as well.
The audience may have questions why the does paper focus on the pruning problem especially for GCN.

W2:  The experiment section needs to be revised. Readers are very easy to get lost.

1)	All the proposed methods and variants should be named. For now is “our GCN baseline”. It is hard for readers to follow.
2)	Some important comparisons are missing. According to the introduction section,
HAN (Liu et al., 2021) is the most related work. However, in the experiments, there is no comparison of “speed up” and “accuracy” between the proposed solution and HAN. Even though these two methods are different, but the experiments still need to show a comparison between the proposed methods and HAN, even with other related methods.
Instead of just showing the performance of proposed method by varying the pruning rate.

3)	The experiment section should introduce all the baselines with more information.

---

### Official Review · Reviewer_h3N6 · 2024-06-12
**Semi-structured cascading pruning masks result in better accuracy and speedup tradeoffs in GCNs for skeleton-based recognition tasks.**

**Confidence:** 3

**Summary:**

The authors utilize a cascading mask-aggregate-selection parametrization to dynamically learn both structured and unstructured pruning masks, resulting in a *semi-structured* approach that combines the accuracy gains from unstructured pruning with the architectural speedups from structured pruning. To achieve this, each weight tensor has a series of masks applied to it. Each mask is a function of the previous (the first takes the weight tensor as the input) and sequentially fulfills one of the three steps:
1. Masking the smallest weights in a traditional unstructured manner
2. Weight-sharing across entries, rows, columns and channels, allowing for semi-structured pruning
3. A gating mechanism which selects block,column/row, or entry wise as the pruning mask for a particular tensor.

They then apply their method to graph convolutional networks specifically on skeleton-based recognition, demonstrating superior results over regularization-based pruning methods.

**Strengths:**

- The paper is generally well-written and is relatively easy to follow.
- The method of semi-structured pruning is interesting and seems novel in this context.
- The proposed method demonstrates impressive results and clearly combines the accuracy strengths of unstructured pruning with the speedup provided by structured pruning.
- Equations and figures are clear and well-formatted.

**Weaknesses:**

- Tables 2 and 3 seem a little unnecessary - the paper focuses specifically on the pruning of GCNs, not on the performance of baseline GCNs themselves, so dedicating half a page to comparing other architectures with GCNs on recognition tasks rather than the proposed pruning method itself is a slightly confusing decision.
- Some sentences run long and could be better worded. (e.g. "Pruning is one of the lightweight network design techniques that operate by removing unnecessary network parts, in a structured or an unstructured manner, including individual weights, neurons or even entire channels." in the abstract).

**Limitations:**

- Given the seemingly general applicability of the proposed method, I question why the authors limited their scope purely to graph convolutional networks and skeleton-based recognition. It is clear that the results demonstrated are impressive, so perhaps it would be insightful to measure the effectiveness of this algorithm across architectures and tasks.

**Suggestions:**

- Utilize the space of Tables 2 and 3 for additional method-specific results.
- In addition to rewording some sentences that run long, there are some small typos throughout the paper. For example, "This allows implementing an annealed (soft) thresholding function that cuts-off all the connections in smooth..." on line 165 is missing an "a" between "in" and "smooth." It will be good to comb the paper over once more to address minor errors like these.

---

### Official Review · Reviewer_Dwcd · 2024-06-13
**A good paper with an interesting approach**

**Confidence:** 3

**Summary:**

In this paper, the authors propose an approach to prune graph convolutional networks (GCNs). Their method involves a 3-stage cascaded pruning mechanism that prunes low magnitude weights while achieving a balance between pruning blocks and pruning individual neurons.
The authors show that their method achieves significant computational speedups relative to the baseline.

**Strengths:**

1. Strong results - their method achieves considerable speedups relative to the baseline and a few competitors when pruning takes place. The baseline network is also more accurate than competitors.

2. Interesting approach - the idea of achieving a balance between pruning individual weights and weight blocks showcases a nice tradeoff between runtime and accuracy.

3. The ablation study is detailed.

**Weaknesses:**

1. It is not clear to me why Equation 7 is an upper bound on the rank. It would be useful to explain this better in the paper.

2. It would be useful to know how much rank-optimization has reduced the rank of the weight matrix compared to the baseline (and perhaps how it changes for other methods in literature). This is not shown in current results.

3. The details of constructing the row and column wise adjacency matrices are not very clear. How are they constructed, and do they stay constant throughout training?

**Limitations:**

Minor weaknesses (as expressed above)

---

### Meta-Review · Area_Chair_jNdA · 2024-06-16

**Recommendation:** Accept (Oral)
**Confidence:** 4

**Metareview:**

## Strengths
* The paper is well written and organized
* The approach is novel and interesting
* The approach provides good performance results by combining the best of both worlds

## Weaknesess
* The paper would benefit from a wider range of experiments (accross larger classes of neural
  networks beyong GCN) to assess the generalizability of the approach
* More comparison with other approaches at the same problem like HAN are needed to assess the
  benefit compared the state-of-the-art solutions

The general sentiment about this paper is rather positive, I recommend acceptance as an oral presentation.

---

### Decision · Program_Chairs · 2024-06-17

**Decision:**

Accept (Poster)

**Comment:**

We thank the authors for their time and contribution to WANT and we are pleased to share that after the reviewing process the paper has been accepted. Congratulations! We encourage the authors to consider reviewers' feedback for the improvement of the camera-ready version. We hope to see you in person at the workshop and brainstorm on efficient training research together!